# Development of the PREMIUM Computerized Adaptive Testing for Measuring the Quality of Information Delivered to Patients with Severe Mental Illnesses

**DOI:** 10.3390/jcm11226687

**Published:** 2022-11-11

**Authors:** Laurent Boyer, Sara Fernandes, Bach Xuan Tran, Guillaume Fond

**Affiliations:** 1School of Medicine-La Timone Medical Campus, EA3279: Health Service Research and Quality of Life Center (CEReSS), Aix-Marseille University, 13005 Marseille, France; 2Fondation FondaMental, 94000 Créteil, France; 3Institute for Preventive Medicine and Public Health, Hanoi Medical University, Hanoi 100000, Vietnam

**Keywords:** information, quality of life, satisfaction, schizophrenia, bipolar disorders, depression

## Abstract

Measuring the quality of information delivered to patients with severe mental illness (SMI), i.e., schizophrenia, bipolar disorders, and major depressive disorders, is essential to improve their quality of care. In this work, we described the different steps of the validation of the PREMIUM computerized adaptive testing (CAT) for measuring the quality of information delivered to patients with SMI. The PREMIUM item bank regarding information included 25 items. A total of 499 patients with schizophrenia (53%), bipolar disorders (26%), and major depressive disorders (22%) were recruited from numerous in- and outpatient settings in France. Unidimensionality, local independence, and monotonicity were verified for 19 items of the item bank. The psychometric properties were satisfactory for both internal (RMSEA = 0.069, CFI = 0.969, TLI = 0.963) and external validity (in particular, significant associations were found with age, educational level, and social functioning). The CAT exhibited satisfactory accuracy and precision (standard error of measurement <0.55 and root mean square error <0.3), with an average administration of eight items. This CAT may be used by healthcare professionals in psychiatric settings to accurately assess the patients’ experience with information.

## 1. Introduction

Schizophrenia, bipolar disorders, and major depressive disorders are severe mental illnesses (SMI) that are characterized by poor quality of care [1,2,3,4]. Previous studies have reported substantial unmet information needs in patients with SMI [5,6]. For example, the need for information has been reported by more than half of patients with schizophrenia in a recent Brazilian study [7]. Sharing information is particularly challenging in patients with SMI because the nature of mental illness is often difficult to explain, and the treatment options and prognosis may vary considerably [8]. Yet, obtaining information about illness, available treatment and care options, prognosis, are strategies for self-management is an essential issue regarding the quality of care because it can affect the process of shared decision making (i.e., “a decision support model allowing information sharing between doctors and patients concerning different care options and taking into account patient preferences”) [9,10] and treatment adherence [11], leading to improved prognosis, including the patients’ quality of life [8]. Poor adherence to treatment may be partly explained by a patient’s belief that the effect of medications is harmful or that medications are unnecessary [12]. It is therefore essential to provide understandable and appropriate information to patients to improve their treatment adherence [13], which remains a major public health problem.

Measuring the quality of information delivered to patients with SMI is an essential step to improve communication strategies for patients with SMI, as well as their quality of care. The French group Patient-Reported Experience Measure for Improving qUality of care in Mental health (PREMIUM) has developed computerized adaptive tests (CATs) for seven domains of quality of care experienced by patients with SMI [14,15]: interpersonal relationships, care environment, drug therapy, access and care coordination, respect and dignity, psychological care, and information. The patient-reported experience measures (PREMs) developed by PREMIUM are expected to optimize measurement precision, to reduce the length of the questionnaire and completion time, and to improve the relevance and the acceptability of these questionnaires in psychiatric settings [16].

In this work, we described the different steps that led to the validation of the PREMIUM CAT for measuring the quality of information delivered to patients with SMI.

## 2. Methodology

### 2.1. Study Setting and Population

We performed a national, multicenter, cross-sectional study between January 2016 and December 2021. Patients were recruited from in- and outpatient psychiatric settings in a French teaching hospital (Assistance Publique-Hôpitaux de Marseille), in the 39 FondaMental academic center of expertise for schizophrenia, bipolar disorders and depression [17] and through social networks, including patients’ associations and with an online web survey. The study protocol was described in more detail elsewhere [14]. All participants gave their informed consent, and the study was carried out in accordance with the Declaration of Helsinki and approved by the local ethics committee (authorization number: 2014-A01152-45).

The inclusion criteria were patients over the age of 18 and under the age of 65, with a DSM-5 diagnosis of schizophrenia, bipolar disorders, or major depressive disorders [18] and having in- or outpatient psychiatric care, informed consent, and speaking/reading French.

### 2.2. Items Measuring the Quality of Information Delivered to Patients with SMI (PREMIUM Item Bank Regarding Information)

The PREMIUM item bank regarding information included 25 items that were self-administered to patients with SMI. The development of the items was detailed in previous works and is based on a mixed method approach associating qualitative and quantitative methodologies [14,15]. All items are presented in Table 1 and scored on a 5-point Likert scale (“strongly disagree”, “disagree”, “neither agree nor disagree”, “agree”, “strongly agree”) with a “not applicable” response option. In addition to these 25 items, we collected an overall satisfaction item (“Overall, were you satisfied with the information about your illness and your management?”) and a visual analogue scale (VAS) (0 minimum to 10 maximum).

### 2.3. Collected Data

The following sociodemographic and clinical data were collected: sex, age, educational level, marital status, occupational status, diagnosis (schizophrenia, bipolar disorders, or major depressive disorders), duration of illness, the Global Assessment of Functioning scale [19] (GAF, ranging from 0 to 100, with a higher score indicating better functioning), quality of life (QoL) using the medical outcome study 12-item Short Form (SF-12) [20], which describes eight dimensions—physical functioning, social functioning, physical role, emotional role, mental health, vitality, bodily pain, general health—and two composite scores for physical and mental quality of life (ranging from 0 to 100, with higher scores indicating a better quality of life).

### 2.4. Statistical Analysis

Descriptive statistics of the sample included frequencies and percentages of categorical variables and the means and standard deviations of continuous variables.

All the items of the PREMIUM item bank regarding information were analyzed using descriptive indicators to select the high performing items.

The item response theory (IRT) assumptions were assessed [21]: unidimensionality using confirmatory factor analysis (CFA) or exploratory factor analysis (EFA), followed by bifactor structure (i.e., one general factor and several group factors), local independence using residual correlations from the CFA model, and monotonicity using visual inspection of characteristic item curves. The generalized partial credit model (GPCM) was used to calibrate the responses to the items [22] after having verified its superiority over other models, such as the partial credit model (PCM) [23]. Item parameters (discrimination and thresholds) were estimated using the maximum marginal likelihood estimation (MMLE) and the expectation–maximization (EM) algorithm [24], and then the model fit to the data was evaluated.

Differential item functioning (DIF) analyses were performed using ordinal logistic regression models [25,26] according to sex (male vs. female), age (median split: patients 40 years or younger vs. patients older than 40 years), care setting (outpatient vs. inpatient), and psychiatric diagnosis (schizophrenia vs. bipolar disorders vs. major depressive disorders). Latent trait scores (θ) were estimated by Bayesian expected a posteriori (EAP) estimation [27]. Then, a linear transformation was performed to obtain θ scores ranging from 0 to 100 (the higher the score was, the better the experience regarding information received). All the indicators of psychometric performance are presented in Appendix A [28,29,30,31,32,33,34,35,36,37,38,39,40,41,42,43,44,45].

Convergent validity was investigated by examining the relationships between item bank scores and psychosocial functioning (GAF), quality of life (SF-12), and satisfaction (overall satisfaction item and VAS). Discriminant validity was investigated by comparing the mean scores of the item bank scores, according to sociodemographic and clinical characteristics, using *t* tests, analysis of variance (ANOVA), and Pearson’s correlation coefficients.

Finally, we developed the item administration algorithm of the CAT. The starting item was the most informative item based on the maximum Fisher information (MFI) criterion [46]. The CAT algorithm then selected the next item with the highest information for the current θ estimate. The CAT algorithm ended when the stopping rule based on the standard error of measurement (SEM) was reached (acceptable range between 0.33 to 0.55 [47]). Three scenarios with different stopping rules corresponding to SEM values of 0.33, 0.44, and 0.55 were simulated and compared to select the best adaptive administration algorithm. The indicators used to assess the performance of the results at each stage of the statistical analyses are provided in Appendix A [28,29,30,31,32,33,34,35,36,37,38,39,40,41,42,43,44,45].

All of the statistical analyses were performed using the following software: IBM PASW SPSS version 20.0 [48], MPlus version 7.0 [49], and R version 4.0.5 [50], using packages “mirt” [51], “lordif” [52], “BifactorIndicesCalculator”, [53] and “mirtCAT” [54].

## 3. Results

### 3.1. Sample Characteristics

A total of 499 patients with schizophrenia (53%), bipolar disorders (26%) and major depressive disorders (22%) were recruited. The characteristics of the participants are presented in Table 2.

### 3.2. Descriptive Indicators of the PREMIUM Item Bank Regarding Information

All the indicators are presented in Appendix A and were satisfactory, except that six items (i.e., I2, I4, I10, I15, I17, and I21) exhibited inter-item correlations that were too high (>0.70), reflecting redundancy between items, and they were thus removed. The lowest scores were reported for item 24 (information about the possibility to access your medical file), item 21 (information about the possibilities of arranging your working time) and item 25 (information about associations and self-help groups.

### 3.3. Assumptions Assessment and Calibration of the IRT Model

Unidimensionality, local independence, and monotonicity were inspected for the remaining 19 items. The fit indices of the 1-factor CFA model were not adequate (RMSEA = 0.126, 95% CI [0.117–0.135], CFI = 0.911, and TLI = 0.900) and were improved with a bifactor structure using one general factor and two group factors (RMSEA = 0.076, 95% CI [0.065–0.086], CFI = 0.975, TLI = 0.968). In the EFA, three factors had eigenvalues greater than 1 (9.0, 1.5, and 1.2, respectively), and the ratio of the first to second eigenvalue was 5.9. The scree plot revealed two predominant factors, while the parallel analysis revealed four predominant factors. All items had higher factor loadings for the general factor than for the group factors. The coefficient ωh for the general factor was 0.93, and those on the two group factors were 0.03 and 0.21, respectively. The percentage of ECV attributable to the general factor was 85.4%, whereas the remaining 14.6% was attributable to the two group factors (7.3% and 10.6%, respectively). All items in the bank were then recoded after examination of the item characteristic curves, which led to an improved model fit (Akaike information criterion, AIC = −8726.88 and Bayes information criterion, BIC = −8886.95). Finally, all residuals were lower than 0.20, and the Cronbach’s alpha was 0.94.

### 3.4. Calibration and Fitting an IRT Model

The GPCM showed a better fit to the data compared to PCM (14,433.67 and 14,539.12 for the AIC and 14,673.79 and 14,703.42 for the BIC, respectively), also supported by the X^2^ = 141.45, *p* < 0.001. In addition, all items showed an adequate fit to the GPCM, with infit values ranging from 0.78 (item I14) to 1.15 (item I1). The item parameters are presented in Appendix A.

### 3.5. Differential Item Functioning

Of the 76 tests performed (i.e., 19 final items of the item bank with 4 confounding factors), 12 exhibited overall DIF but with negligible magnitudes: 5 items for sex (items 3, 16, 19, 22, and 24), 1 item for age (item 7), 5 items for care setting (items 7, 16, 18, 19, and 22) and 5 items for diagnosis (items 5, 12, 20, 22, and 25). All the DIF results are presented in Appendix A.

The test information curve of the item bank confirmed that the items have a high measurement precision over a broad spectrum of the latent trait (61.0% of total information is included in the [−2,2] range of the latent continuum values) (Figure 1). Item 13 was the most informative of the bank—“the course of your care (how to take your drug therapy, the frequency of your appointments, etc.”, whereas item 20 was the least informative—“existing medical-social aid (reimbursement of health care costs, access to home help, obtaining the disabled adult allowance, etc.”).

### 3.6. External Validity of the PREMIUM Item Bank Regarding Information

The mean score was 46.97 ± 19.82. The scores were strongly correlated with overall satisfaction and the corresponding VAS, confirming the convergent validity. All the SF-12 dimensions were significantly correlated with information experience, but the relationships were stronger regarding social and emotional functioning.

Concerning discriminant validity, younger age, higher educational level, and online survey were associated with poorer information experience. No significant differences were found by sex, marital status, employment status, care setting, diagnosis, duration of illness, and psychosocial functioning. The results regarding the external validation are presented in Table 3.

### 3.7. Elaboration of the Item Administration Algorithm

Among the three scenarios tested, the CAT simulation with a level of precision of SEM < 0.33 was the most efficient, with the highest levels of accuracy (r = 0.97) and precision (RMSE = 0.23) while administering eight items, on average. Table 4 provides the results of the CAT simulations.

## 4. Discussion

In this study, we demonstrated the validity of the PREMIUM-CAT for measuring the quality of information delivered to patients with SMI. This CAT presents interesting characteristics for widespread use in psychiatric settings.

The PREMIUM item bank has been developed based on a literature review and patient interviews, ensuring its content validity [15,55]. Similar to a previous study on information needs in mental health [56], the PREMIUM item bank regarding information explores a broad facet of the information delivered to patients, including: basic facts (e.g., diagnosis, symptoms, etc.), treatment (e.g., side effects, etc.), coping (e.g., lifestyle advice, etc.), medical systems (e.g., possibility of accessing medical file, etc.), social aids, professional accommodations, and peer support. The basic facts are the information needs most frequently reported in the literature in contrast to self-help and peer support [56]. These findings are corroborated by our study in which the item regarding the course of care (I13) was the most informative of the bank, while the item on social aids (I20) was the least informative of the bank. Measurement invariance was demonstrated by the few indications of negligible magnitude DIF, which implies that the psychometric properties of the items are invariant across gender, age, care setting, and diagnosis, and that the item bank scores can be compared between respondents, regardless of their characteristics. The different sampling strategies may also have impacted participants’ information experience at different levels. Response bias is known to affect self-reported data, leading to results biased in a positive direction, as in traditional satisfaction surveys [57]. Participants recruited from healthcare facilities are more likely to provide biased responses due to concerns about the impact on their care and relationships with the healthcare team. Participants recruited from healthcare were invited to participate in the study by a psychologist external to the healthcare team in order to limit this bias. However, the scores were significantly higher for participants from the healthcare facilities than for participants from the online survey. External validity, explored by the use of socio-demographic characteristics and established psychiatric and functioning measures, globally confirmed our assumptions. In particular, we confirmed that information experience was slightly and positively correlated to quality of life, which has been shown in other chronic severe illnesses, such as cancer [58]. Conversely, our results showed no statistical difference in information experience according to psychosocial functioning, even though most participants had poor psychosocial functioning. Our results also showed that participants with high educational level and young age reported a poorer information experience. Patients’ preference about information may vary according to age and education [59]. Health literacy and internet use may also explain these results, in part [60,61,62,63]. Innovative information strategies could be implemented in novel digital adherence-assessment devices [64] and telemedicine interventions [65]. Finally, the item bank offers better precision and accuracy measurement over the [−2;2] range of theta values, while reducing response burden on participants, with an average of 8 items administered.

The analysis of the item scores has yielded three points of information that should be targeted with high priority. First, participants reported poor information experience on the right to access her/his medical file. All physicians are informed during their training of the 2003 Kouchner Law obliging institutions to provide medical records within 8 days to any patient on demand. The reasons for this poorer experience should be explored in future studies. However, it seems necessary to reinforce the information of this right to all patients, perhaps by automatic mailing or phone texting, for instance. Second, participants reported poor information experience regarding social aids and professional accommodations. This information is not directly related to medical care, but rather to medico-social/social/rehabilitation care and occupational medicine. Poor access to medico-social and social care and lack of knowledge regarding these services may explain why patients report poorer information experience regarding this information. All patients could be delivered systematic written information regarding social aids and professional accommodations when receiving her/his first diagnosis. Third, participants reported poorer information experience regarding associations or programs aimed at helping patients. This poor experience may be due to a lack of knowledge. Physicians are themselves poorly informed about all associations, programs, and initiatives concerning all mental illnesses. This information also may vary across time and geographical areas. Caregivers may also estimate that their role is limited to medical care. Future qualitative studies should explore the reasons obehind this poorer experience and how to address this issue.

### 4.1. Futures Directions

The item bank and its CAT version provide a brief, accurate, and flexible assessment of the information experience of patients with SMI, suitable for research and clinical practice. The systematic and continuous collection of PREMs through a digital platform has great potential to inform quality improvement actions in mental health care [66]. However, the CAT version requires an appropriate information technology (IT) infrastructure to collect data, at the hospital and/or at home, requiring that patients have internet access. Future dissemination and implementation work will need to determine the feasibility of such a device for routine use and its effectiveness in meeting specific research or clinical needs.

### 4.2. Limitations

Our sample size is large compared to traditional psychiatric studies, but may be limited in the field of psychometric studies. However, our sample size was sufficient, since sample size requirements recommend approximately 500 observations for accurate parameter estimates with multiparameter models (e.g., GPCM) [67,68]. In addition, the sample included a diverse patient population, both inpatient and outpatient, from several facilities in different geographic regions of the country. Future work with a larger and more diverse sample will allow for additional psychometric testing to develop a better understanding of the metric properties, but also to examine the generalizability of this item bank to other populations, such as patients with common mental disorders (e.g., anxiety disorders). Second, in this study, as in previous PREMIUM studies [12,69], we used the GPCM, which showed an adequate fit to the data and whose fit can be compared to the PCM because these are nested models. The graded response model (GRM), which tends to provide similar results, could have been an alternative (e.g., in case of non-convergence) [70]. Third, criterion validity could not be assessed because, to the best of our knowledge, no gold standard was available. In addition, given the cross-sectional study design, responsiveness could not be studied. Fourth, CAT precision depends on the quality of the related item bank, which showed the highest measurement precision in the [−2;2] range of theta values. Future studies should evaluate whether additional items are needed to cover the ends of the latent continuum. Fifth, we did not evaluate health literacy, which may have influenced information experience [71]. Likewise, symptom severity was not measured using psychiatric scales (e.g., PANSS, MADRS, YMRS) due to diagnostic heterogeneity and different recruitment strategies that did not allow for hetero-assessment. By using the GAF, which provides an approximate measure of the severity of these symptoms, we examined the impact of the severity of mental illness on the information experience. Future studies should address these questions. Finally, the precision and accuracy of the CAT should be re-evaluated in an independent sample and under real-world conditions to assess whether the responses may be affected by other factors [72].

## 5. Conclusions

The PREMIUM item bank regarding information and its CAT version may be used by healthcare professionals in psychiatric settings to accurately assess the experience of patients with information. Particular attention should be paid to informing patients about their right to access their medical files, social aids, professional accommodations, and peer support services.

## Figures and Tables

**Figure 1 jcm-11-06687-f001:**
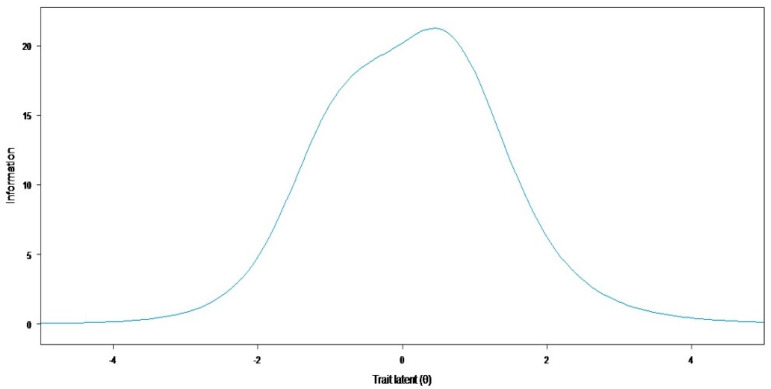
Test information.

**Table 1 jcm-11-06687-t001:** List of items. You have been sufficiently informed about.

Item No.	Content Item
I1	the symptoms of your illness
I2	the possible evolution of your illness
I3	the impact of your illness on your daily life
I4	the services and structures relevant to managing your illness (e.g., rehabilitation programs, etc.)
I5	the existence of initiatives or programs designed to help you learn about, understand, and cope with your illness much more effectively
I6	the reasons for your treatments (medication, psychotherapy, etc.) and your care (consultation, hospitalization, etc.)
I7	how your treatments (medication, psychotherapy, etc.) and your care (consultation, hospitalization, etc.) can help you
I8	your general health, with advice and recommendations on your diet, sleep, physical activity, etc.
I9	what to do if your treatments and care do not seem to be working
I10	the need to take your treatment, even if it does not seem effective
I11	the need to take your treatment, even if you feel better
I12	the side effects and repercussions of your treatments and care on your health (difficulty in concentrating, feeling tired, constipation, weight gain, etc.)
I13	the course of your care (how to take your drug therapy, the frequency of your appointments, etc.)
I14	the reasons why your exams (physical, blood, questionnaire completion, imaging, etc.) were prescribed
I15	the course of the exams (physical, blood, questionnaire completion, imaging, etc.) that you have performed
I16	the results of exams (physical, blood tests, questionnaire completion, imaging, etc.) that you have performed
I17	how to get help in an emergency
I18	who to contact in an emergency
I19	the possibility that you will be hospitalized if your health status requires it (in what circumstances, the modalities of admission)
I20	existing medical-social aid (reimbursement of health care costs, access to home help, obtaining the disabled adult allowance, etc.)
I21	the possibilities of arranging your working time
I22	the possibilities of accommodation that could be useful in your daily life (at home, at work, etc.)
I23	the possibility of refusing some care that is proposed to you
I24	the possibility to access your medical file
I25	about associations and self-help groups (discussion groups, patient associations, etc.) from which you can obtain information about your illness and meet people who have experienced the same situation as you

**Table 2 jcm-11-06687-t002:** Sample description (N = 499).

	Effective (Percentage) or Mean ± Standard Deviation
**Sociodemographic data**	
Study participation (*n* = 499)	
Healthcare settings	247 (49.5)
Online survey	252 (50.5)
Sex (male) (*n* = 499)	255 (51.1)
Age, years (*n* = 498)	40.49 ±11.70
Marital status (single) (*n* = 478)	355 (74.3)
Educational level (<bachelor’s degree) (*n* = 480)	134 (27.9)
Employment status (unemployed) (*n* = 478)	353 (73.8)
**Clinical data**	
Care setting (*n* = 499)	
Outpatient	399 (80.0)
Inpatient	100 (20.0)
Involuntary commitment	37 (37.1)
Psychiatric diagnosis (*n* = 484)	
Schizophrenia	255 (52.7)
Bipolar disorders	124 (25.6)
Major depressive disorders	105 (21.7)
Illness duration, years (*n* = 472)	13.34 ± 9.23
<5 years	99 (21.0)
≥5 years	373 (79)
Global Assessment of Functioning (GAF) (*n* = 216)	60.52 ± 15.64
Poor functioning (<61)	135 (62.5)
Good functioning (≥61)	81 (37.5)
Quality of life (SF-12 score)	
Physical functioning (*n* = 491)	46.46 ± 11.35
Social functioning (*n* = 492)	33.87 ± 11.76
Role physical (*n* = 492)	40.40 ± 11.17
Role emotional (*n* = 492)	33.00 ± 12.50
Mental health (*n* = 494)	44.91 ± 11.08
Vitality (*n* = 492)	50.96 ± 10.24
Bodily pain (*n* = 494)	44.05 ± 12.77
General health (*n* = 493)	34.85 ± 10.70
Physical composite quality of life score (*n* = 485)	43.85 ± 10.30
Mental composite quality of life score (*n* = 485)	38.92 ± 11.55

**Table 3 jcm-11-06687-t003:** External validity.

	Correlation Coefficient(r)	Mean ± Standard Deviation	*p*-Value
Sociodemographic data			
Study participation	-		<0.001
Healthcare settings		51.81 ± 18.90	
Online survey		42.23 ± 19.60	
Age	0.22	-	<0.001
Sex	-		0.096
Male		45.57 ± 21.43	
Female		48.55 ± 18.14	
Marital status	-		0.27
Single		46.35 ± 19.58	
Non-single		48.64 ± 20.43	
Educational level	-		0.007
<Bachelor’s degree		50.94 ± 20.54	
≥Bachelor’s degree		45.44 ± 19.47	
Employment status	-		0.425
Employed		48.18 ± 19.71	
Unemployed		46.52 ± 20.04	
**Clinical data**			
Care setting	-		0.74
Outpatient		47.14 ± 20.25	
Inpatient		46.38 ± 18.09	
Psychiatric diagnosis	-		0.126
Schizophrenia		45.52 ± 18.45	
Bipolar disorders		49.92 ± 21.44	
Major depressive disorders		46.57 ± 20.87	
Duration of illness			0.134
<5 years		49.57 ± 16.82	
≥5 years		46.21 ± 20.52	
Global Assessment of Functioning (GAF)			0.333
Poor functioning (<61)		50.62 ± 18.75	
Good functioning (≥61)		53.14 ± 18.16	
**Proxy measures**			
Item of overall satisfaction	0.41	-	<0.001
Visual analogue scale	0.4	-	<0.001
Quality of life (SF-12)			
Physical functioning	0.14	-	0.002
Social functioning	0.27	-	<0.001
Physical role	0.23	-	<0.001
Emotional role	0.26	-	<0.001
Mental health	0.2	-	<0.001
Vitality	0.12	-	0.009
Bodily pain	0.11	-	0.011
General health	0.22	-	<0.001
Physical composite quality of life score	0.14	-	0.002
Mental composite quality of life score	0.25	-	<0.001

**Table 4 jcm-11-06687-t004:** Mean scores and precision indicators for each CAT simulation.

Precision Level	Indicators	
**SEM < 0.33**	Mean score	46.89 ± 19.15
	Correlation coefficient (r)	0.97
	RMSE	0.23
	Mean number of items	8.13
**SEM < 0.44**	Mean score	46.84 ± 18.27
	Correlation coefficient (r)	0.94
	RMSE	0.34
	Mean number of items	4.35
**SEM < 0.55**	Mean score	50.29 ± 24.08
	Correlation coefficient (r)	0.90
	RMSE	0.43
	Mean number of items	2.43

**Abbreviations**: SEM, standard error of measurement; RMSE root mean square error.

## Data Availability

The data are available on demand from the PREMIUM Scientific Committee.

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
