# Peer review of "Development of the PREMIUM Computerized Adaptive Testing for Measuring the Quality of Information Delivered to Patients with Severe Mental Illnesses"

_jcm, 2022, doi:10.3390/jcm11226687_

Round 1
Reviewer 1 Report
The present study aimed to describe the different steps that led to the validation of the PRE- 50 MIUM CAT for measuring the quality of information delivered to patients with severe mental disorders. The work is well-performed, well written and it is a relevant topic of research.
I just have three comments to contribute to improve the discussion.
1) What was the impact of sample strategy recruitment on results?
2) Do you think that mood or psychotic symptoms could have influenced the results?
3) Most of the patients had poor psychosocial functioning. What would be the influence of this factor on results?
4) Usually, the validity and reliability of new instruments used to be compared with a gold standard.
5) How generalizable are the present results?
6) What is the real relevance of this new instrument to the research and clinical practice?
Reviewer 2 Report
In this manuscript the authors described the validation of the Computerised Adaptive Tests (CATs) for measuring quality of information delivered to patients suffering from Severe Mental Disorders (SMDs).
The topic is of great novelty and interest and perfectly fits the Special Issue, to which it may provide a valid contribution.
The design of the study and the methods are expressed clearly and answer perfectly to its aim. The discussion is supported by the results and the interpretation of the data leads to interesting conclusions.
Following a careful reading of your work, I recommend the following suggestions:
- Introduction: please include a more relevant reference (ex: WHO reports) regarding the unmet needs and lack of quality of care of people suffering from SMDs (page 1, line 28-32)
Concerning the interesting topic of “shared-decision making” (page 1, line 36-38) I would suggest to add a more relevant reference (ex: Thomas EC, Ben-David S, Treichler E, et al. A Systematic Review of Shared Decision-Making Interventions for Service Users With Serious Mental Illnesses: State of the Science and Future Directions. Psychiatr Serv. 2021;72(11):1288-1300.)
The topic of intervention to increase adherence (page 1, line 39) to treatment is of great interest, I wonder if the authors may explore it more in depth and add some other relevant references (i.e. Kikkert MJ, Schene AH, Koeter MW, et al. Medication adherence in schizophrenia: exploring patients', carers' and professionals' views. Schizophr Bull. 2006;32(4):786-794.)
- Results: I suggest some graphical editing for both the presented Tables and the Supplementary material in order to make the results more understandable.
In particular the common reader may find it difficult to interpretate the results from Supplementary Table 2 and Supplementary Table 3 while reading the manuscript.
Please include footnotes by specifying the acronyms or abbreviations.
- Discussion: I really appreciated this section and the interpretation of data the authors provided.
Regarding the possible reasons behind the difference in age and education (page 8, line 219), I suggest the authors may mention some digital health interventions that have been put into practice to increase adherence (ex: Papola D, Gastaldon C, Ostuzzi G. Can a digital medicine system improve adherence to antipsychotic treatment?. Epidemiol Psychiatr Sci. 2018;27(3):227-229; Basit SA, Mathews N, Kunik ME. Telemedicine interventions for medication adherence in mental illness: A systematic review. Gen Hosp Psychiatry. 2020;62:28-36. doi:10.1016/j.genhosppsych.2019.11.004)
- Limitations: please, support the statement “the sample size was large enough to calibrate the item pool” (page 8, line 243-244).
Please, explore the other possible models beside GPCM (page 8, line 248).
I understand that due to the heterogeneity of the diagnosis this would have been difficult, but the authors should mention the lack of severity indexes (ex: PANSS, MADRS, YMRS), although the setting of care may offer a sort of proxy measure.
- Finally, I would suggest a further English editing as there are some points with grammatical errors.
